# How Italy Tweeted about COVID-19: Detecting Reactions to the Pandemic from Social Media

**DOI:** 10.3390/ijerph19137785

**Published:** 2022-06-24

**Authors:** Valentina Lorenzoni, Gianni Andreozzi, Andrea Bazzani, Virginia Casigliani, Salvatore Pirri, Lara Tavoschi, Giuseppe Turchetti

**Affiliations:** 1Institute of Management and Department of Excellence EMbeDS, Scuola Superiore Sant’Anna, 56127 Pisa, Italy; gianni.andreozzi@gmail.com (G.A.); andrea.bazzani@santannapisa.it (A.B.); salvatore.pirri@santannapisa.it (S.P.); giuseppe.turchetti@santannapisa.it (G.T.); 2Department of Translational Research and New Technologies in Medicine and Surgery, University of Pisa, 56126 Pisa, Italy; v.casigliani7@gmail.com (V.C.); lara.tavoschi@unipi.it (L.T.)

**Keywords:** COVID-19, social media, Twitter, health emergency, emotional and behavioural reaction, emergency management

## Abstract

The COVID-19 pandemic required communities throughout the world to deal with unknown threats. Using Twitter data, this study aimed to detect reactions to the outbreak in Italy and to evaluate the relationship between measures derived from social media (SM) with both national epidemiological data and reports on the violations of the restrictions. The dynamics of time-series about tweets counts, emotions expressed, and themes discussed were evaluated using Italian posts regarding COVID-19 from 25 February to 4 May 2020. Considering 4,988,255 tweets, results highlight that emotions changed significantly over time with anger, disgust, fear, and sadness showing a downward trend, while joy, trust, anticipation, and surprise increased. The trend of emotions correlated significantly with national variation in confirmed cases and reports on the violations of restrictive measures. The study highlights the potential of using SM to assess emotional and behavioural reactions, delineating their possible contribution to the establishment of a decision management system during emergencies.

## 1. Introduction

On 31 December 2019, the World Health Organization (WHO) was informed by the Chinese authorities of an outbreak of pneumonia of unknown cause in the city of Wuhan [1]. Two months later, the coronavirus (COVID-19) responsible for the cases of pneumonia—officially named severe acute respiratory syndrome coronavirus 2 (SARS-CoV-2) by the WHO in February 2020—had already spread all over the world, thus becoming a global health emergency. The pandemic affected communities throughout the world, with short- and medium-term consequences not only on health (physical and mental) but also on society as a whole.

Similarly to previous epidemics, a wave of emotions and behaviours swept over communities in parallel with the diffusion of the virus [2,3,4]. The virus also went viral on social media (SM). A month later, when WHO declared the SARS-CoV-2 to be a global public health emergency [5], billions of users around the world had already started to share news, opinions, and emotions on social networks regarding the consequences of this new coronavirus outbreak [6,7].

In the digital era, SM represent the easiest tool to keep oneself informed. During the COVID-19 outbreak, worldwide health agencies, institutions, and scientific journals regularly published information through a large number of platforms, reaching millions of people [8].

A survey conducted during the first month of the cordon sanitaire in Wuhan and another study in Taiwan showed that the majority of participants interviewed sought COVID-19 information on SM or online [9,10].

Due to the increasing popularity of SM platforms, researchers have already used them to track the emotional and sentimental response to emergencies during previous epidemics or disasters (e.g., Zika, Ebola, and H1N1 epidemics as well as the Fukushima disaster). While a branch of literature has focused on the use of SM to estimate the number of cases during epidemics [11,12,13], few studies have explored the use of big data from SM to analyze the themes most frequently discussed during the emergency, as well as the related emotions elicited and their link to users’ psychological responses and social behaviour [14,15,16,17].

During emergencies, policymakers are often faced with the problem of ineffective or misleading communication, which can generate a state of panic and confusion in the population. This highlights the importance of not only correctly managing communication during emergencies but also monitoring its impact on specific audiences.

Users using SM platforms such as Twitter form virtual communities that discuss their life and various topics through the use of likes, hashtags, mentions, and retweets that form a complex network [18].

Social media are also becoming more and more used as an alternative source for evidence-based information on health, thus having the potential of representing an additional valuable source for surveillance of public health, which can affect the impact of the intervention as well as the engagement of right and wrong health-related behaviour [19,20].

In this context, the use of data from SM represents an opportunity to develop a real-time surveillance system, providing policymakers with a continuous and timely assessment of people’s behaviours and people’s interpretation of public health intervention, together with proxy measures of social awareness, risk perception and compliance with recommendations [14,15,21]. 

Based on the real-time feedback from a similar system, it will be possible to immediately detect “problems” and also to direct citizens to appropriate sources of information—counteracting misinformation [22] and enhancing risk communication—aimed at helping in the diagnosis of a disease and in the referral to appropriate health care services, including psychological support and progress in terms of scientific research [8,23]. 

As the first country in the West that was heavily hit by the pandemic, Italy was also the first to introduce strict measures of containment and isolation [24]. In Italy, from the first confirmed case of community transmission (21 February) to the beginning of the lockdown (9 March), the number of infected cases rapidly increased to 9172, with 463 deaths [25]. 

The quarantine already in place in the regions of Lombardy and Veneto at the end of February was then extended to the rest of the country [26]. The measures imposed by the nationwide lockdown dramatically limited most working and social activities. It was also unprecedented, and triggered contrasting behavioural and emotional responses, with potential short- and long-term psychological and social consequences. 

Although researchers analyzed SM in order to understand how different populations and countries were affected by the COVID-19 emergency [7,27,28], to the best of our knowledge, there are few studies on the use of SM to understand public reactions to the COVID-19 pandemic in Italy. Neither are there studies providing a comprehensive overview of the dynamics of information and emotions shared over such an extensive time period and evaluating the relationship between measures derived from SM with both national epidemiological data and reports on the violations of the restrictions.

This study aimed to track the reactions on Twitter in response to the news of the COVID-19 pandemic in Italy during a period of over two months of restrictive measures, from 25 February to 4 May 2020. The focus is on the dynamics of the volume of tweets, and the trends of their emotional and semantic content. The final aim was to assess the utility of the data from SM, in this case, Twitter, for the continuous monitoring of people’s emotional and behavioural reactions during a health emergency, and for the detection of emerging themes.

## 2. Materials and Methods

*Setting.* Italy was the first country in Europe to be affected by the COVID-19 pandemic. After the detection of imported cases, the first locally acquired cases were detected in mid-February 2020. On 23 February 2020, the threshold of 100 cases was exceeded. Accordingly, on 23 February, the first-decree law was enacted, which imposed restrictive measures on some municipalities in the north of Italy; restrictive measures were then extended to six other regions on 25 February 2020 [29]. On 9 March, the government imposed a national quarantine [26], which was then extended until 4 May 2020 [30] and gradually lifted afterwards. 

*Data sources and pre-processing.* A descriptive study was performed using Twitter data and data from the Italian Department of Civil Protection. Indeed, using the rtweet package [31] in R software, tweets posted in Italian between 25 February 2020 and 4 May 2020 and related to conversations about COVID-19 were retrieved daily considering specific trending hashtags (i.e., #COVID19, #COVID_19, #COVID2019, #COVID, #iorestoacasa, #coronavirus, #coronavirusitalia, #COVID-9, #lockdown) as keywords for data extrapolation.

After removing duplicates, the remaining sample, a total of 4,988,255 Twitter posts (1,256,890 tweets and 3,731,365 retweets) referring to conversations about COVID-19 shared in Italian during the study time frame, constituted the study database and was then pre-processed (i.e., normalization and cleaning of the text of the tweet, tokenization of the words) using the tidytext R package [32]. 

All analyses used public, anonymized data adhering to the terms and conditions, terms of use and privacy policies of Twitter.

The Italian dataset of COVID-19 data (i.e., total hospitalized, home isolation, total positives, total variation in newly confirmed cases, new positives, discharged healed, deceased, total cases) per region throughout the time frame of interest (25 February–4 May) was built using daily data available from the Italian Department of Civil Protection [25]. Data on the number of daily violations of restriction measures issued were also collected [33].

*Emotion analysis.* Since in situations such as the health emergency context considered in the present study, there is typically an overexpression of negative sentiment on SM that may produce noisy estimates when classifying sentiment, emotions were instead evaluated. Using the get_nrc_sentiment function implemented in the syuzhet R package [34], emotions expressed in each tweet were identified and scored according to Saif Mohammad’s National Research Council (NRC) Emotion Lexicon [35]. Basically, the NRC associates the retrieved text words with eight emotions: anger, fear, anticipation, trust, surprise, sadness, joy, and disgust. For each emotion, the daily (and weekly) proportion of the number of words (used in tweets) expressing that emotion was obtained by dividing the number of words related to the specific emotion by the total number of words expressing positive and negative sentiment used in tweets that day. A similar process was performed for the weekly proportion of tweets expressing the various emotions.

*Semantic analysis*. To explore the most common themes in the tweets, a semantic analysis of the most frequent pairs of adjacent words, bigrams (such as “primo caso” or “porte chiuse”), was performed. For each of the 10 weeks covering the analysis period, the most frequent 50 bigrams of words were extracted. After removing the bigrams formed by two verbs, two coders independently reviewed the bigrams and assigned them to specific categories identified in a work about COVID-19 in the US [28] in accordance with their semantic meaning. A third coder solved disagreements, analyzing independently the bigrams and discussing the coding with the other two coders to decide the category. Then, the two main coders reviewed the categories assigned and aggregated them into six major themes: “virus and pandemic impact”, “geopolitics and government”, “medical support”, “preventive measures”, “information seeking”, and “community” (Appendix A). Furthermore, an additional theme, “time”, was identified.

Accordingly, seven themes were thus used, and the number of bigrams and their relative frequency were recorded. Results from the semantic analysis considering bigrams and individual words are similar, and for the sake of simplicity, just the bigrams were recorded.

*Statistical analysis.* Statistical analyses performed aimed at describing and detecting eventual patterns in time-series related to the volume of tweets posted, the percentage of tweets expressing specific emotions, and the frequency of mention of themes identified through the semantic analysis to both evaluate the dynamic of the public’s reactions to the pandemic over time and also to assess relationships between those series. Moreover, also considering and evaluating time-series related to epidemiological data (i.e., daily series of the variation in the number of new cases) as well as of violations of the restrictive measures analyses aimed also at identifying possible relationships between them and data from Twitter. 

In detail, in order to explore the dynamics of the volume of tweets throughout the time period considered, the daily number of tweets was calculated, also differentiating verified (accounts considered to be of public interest, such as institutions, politicians, famous people, etc.) from non-verified accounts. The average weekly number of tweets over the 10 weeks of analysis was also obtained. For both the volume of posts and the percentage of posts expressing specific emotions, original daily time-series and series obtained using the simple moving average (MA) based on 7-day points were generated to identify patterns over time, smoothing out peaks and the “weekend effect” (i.e., a drop in the number of tweets on Saturdays and Sundays). Indeed, MA is a method used to smooth out short-term fluctuations in time-series data to highlight long-term trends or cycles, and it is obtained by averaging *k* data points of the original time-series, where k is the order of the MA, *k* = 7 in the present analysis. 

The non-parametric Mann–Kendall trend test was used to explore the presence of a monotonic trend in the time-series considered. Given a time-series evaluated over n data points, the Mann–Kendall test is based on the following statistics:$$S=\sum_{i=1}^{n-1}\overset{n}{\underset{j=k+1}{\sum }}sign(x_{j}-x_{i})$$

Spearman rho correlation was used to assess the relationship between the mean weekly frequency of emotions and the themes tweeted (identified through the semantic analysis).

Finally, the cross-correlation function was derived to assess correlations between time-series related to the percentage of tweets expressing specific emotions and epidemiological data (i.e., daily series of the variation in the number of new cases) as well as of violations of the restrictive measures.

All analyses were performed using R version 3.6.2, and *p*-value < 0.05 was considered for statistical significance.

## 3. Results

### 3.1. Volume of Twitter Posts

Figure 1 reports the daily counts of tweets (and retweets) over the study period with key dates/events highlighted together with the daily count of new cases detected and the cumulative number of cases. The number of posts published peaked around 9 March 2020 when the President of the Council of Ministers imposed a decree ordering a national quarantine, and thereafter the daily count of tweets fluctuated. Considering the time-series related to the daily variation in newly confirmed cases, no overlap with the daily counts of tweets was found.

Differentiating between verified and non-verified accounts, it became evident that activities of the Twitter community were largely driven by verified accounts; in fact, despite the difference in terms of volume, the timeline of tweet counts shared by the two types of accounts overlapped (Figure 2).

During the study period, the mean daily number of posts increased from 21,720 in week 1, tripling in week 2, and doubling again in week 3 when the national lockdown was imposed. The mean number of posts slightly decreased in weeks 4 and 5, with around 50,000 to 60,000 posts per day in the following weeks (Appendix A).

### 3.2. Emotion Analysis

Trust was the most commonly reported emotion in the tweets retrieved (39.4%), followed by fear (34.1%), and anticipation (30%). The remaining emotions (i.e., joy, sadness, anger, surprise, disgust) were detected in less than 30% of tweets even though the data varied considerably over the study period (Figure 3).

The trend analysis showed a significant decrease over time in the frequency of tweets expressing anger (tau = −0.161, *p*-value = 0.049), disgust (tau = −0.314, *p*-value *<* 0.001), fear (tau = −0.499, *p*-value *<* 0.001), and sadness (tau = −0.345, *p*-value < 0.001); conversely, there was a significant increase in the percentage of tweets expressing anticipation (tau = 0.212, *p*-value = 0.010), joy (tau = 0.408, *p*-value *<* 0.001), surprise (tau = 0.215, *p*-value = 0.009) and trust (tau = 0.313, *p*-value < 0.001).

### 3.3. Word Frequency and Semantic Analysis

Table 1 reports the frequency bigrams detected in tweets posted throughout the study period according to the seven themes defined by the sematic analysis. Top bigrams associated with each theme are also reported.

As a whole, “Preventive measures” was the most frequently discussed theme, followed by, in order of citation (from the most frequent to the least frequent): “Geopolitics and government”, “Medical support”, “Virus and pandemic impact”, “Community”, “Information seeking”, and “Time”.

However, the frequency of the themes identified varied slightly over time (Figure 4).

Themes were identified through semantic analysis of the 50 most frequent bigrams in each week. The weekly percentage citation frequency was then obtained by dividing the occurrence of bigrams by the overall frequency of mentions associated with the bigrams extracted.

In particular, a significantly increasing trend was found for “information seeking” (tau = 0.644, *p*-value = 0.012), while a significant decreasing trend was detected for “preventive measure” (tau = −0.556, *p*-value = 0.032). No significant trend was found for the other themes. Of note, while bigrams related to “virus and pandemic impact” peaked in the first week (corresponding to the initial phase of the epidemic), most themes, particularly “geopolitics and government”, fluctuated over time. Bigrams related to “preventive measures” peaked in the third week, i.e., when the national lockdown was imposed, and decreased thereafter.

Pairwise correlations between the themes suggested a significant positive correlation between bigrams related to “information seeking” with both “geopolitics and government” and “virus and pandemic impact” (rho = 0.490, *p*-value = 0.048 and rho = 0.590, *p*-value = 0.042, respectively). On the other hand, the frequency of bigrams related to “preventive measures” correlated negatively with both “geopolitics and government” and “information seeking” (rho = −0.840, *p*-value = 0.001 and rho = −0.670, *p*-value = 0.022, respectively); a significant negative correlation was also found between the use of words related to “time” and “geopolitics and government” (rho = −0.590, *p*-value = 0.031); see Appendix A.

### 3.4. Correlations between Emotions and Themes Discussed

The correlation between the frequency of themes identified and the percentage of tweets expressing a specific emotion on a weekly base (Appendix A) revealed a significant negative correlation between “preventive measures” and both joy (rho = −0.724, *p*-value = 0.018) and trust (rho = −0.879, *p*-value = 0.002). The weekly percentage of tweets expressing surprise correlated positively with the tweeting of bigrams related to “medical support”. Combinations of words related to “information seeking” correlated positively with joy (rho = 0.669, *p*-value = 0.034), while they showed a negative correlation with fear (rho = −0.650, *p*-value = 0.042).

*Correlations between emotions, epidemiological data, and reported violations*. The cross-correlation functions between the daily time-series referring to the percentage of tweets expressing specific emotions and those related to variations in the number of new cases was derived. With regard to negative time lags, as the number of new cases increased, the percentage of tweets expressing trust significantly decreased. Anticipation followed a similar trend, and the correlation was significant for both positive and negative lags (Appendix A).

On the other hand, for positive lags, a significant positive correlation was found between the variations in the number of new cases and the percentage of tweets expressing sadness.

With respect to the cross-correlation function between the percentage of tweets associated with the different emotions and the percentage of subjects fined for violation of restrictions, a significant positive correlation was found with anger (both positive and negative lags), fear (positive lags), and sadness (positive lags), while there was a significant negative correlation with trust (Appendix A).

## 4. Discussion

The overall aim of this study was to understand how Twitter users in Italy reacted to the COVID-19 outbreak and how the control measures implemented in the country to combat the spread of the virus triggered specific emotions.

The hypothesis motivating the present study was that during emergency situations Twitter might be a valuable “social awareness stream” for policymakers as a proxy for people’s emotions (e.g., reaction to control measures) and reactions (e.g., adherence to restrictive norms). Indeed, the initial phase of a public health emergency is one of the most critical from a communication perspective, and that motivated the choice of the time frame for the analysis. 

Given the current widespread use of SM [36], the appearance of COVID-19-related posts on SM platforms was much higher than in previous epidemics [37], and many public health departments and authorities used SM platforms to communicate and share information about the emergency [38]. 

For those reasons, these digital platforms and the general web-based activity related to COVID-19 have been investigated to define risk communication strategies and assess the public response [39,40]. Despite that, studies on Western countries have been limited, and only a few have provided a comprehensive overview of the potential of exploiting SM data to understand public reactions, considering an adequate time frame across the various phases of the epidemic [41,42]. 

The main results of the present study are that (1) the volume of conversations on Twitter is essentially related to key events (i.e., the beginning of the lockdown) occurring at a national level, rather than to the epidemiological trend; (2) emotions and themes expressed in tweets changed during the study period; (3) emotions and themes were intertwined; and (4) the dynamics of the emotions correlated with the national epidemiological trend and the number of subjects violating the restrictive measures imposed.

The epidemic initially impacted on activities on SM platforms [6], stimulating a large volume of discussion about COVID-19. Previous evidence shows diverging results on the relationship between the volume of activity on Twitter, which is probably related also to the difference in Twitter usage in diverse contexts, and epidemiological trend. Indeed, while a branch of evidence showed that differences in the volume of Twitter posts between countries reflected differences in the incidence of COVID-19 cases [7], other studies found that people’s interest in infectious diseases on SM was rather associated with the latest news and key events [6,43]. The latter findings are in line with results from the present study showing the number of tweets about COVID-19 within the Italian Twitter community was essentially driven by key events (i.e., the announcement of the national lockdown, the WHO’s declaration of pandemic) and by the posts published by verified users (i.e., public institutions or popular accounts), rather than by the epidemiologic course of the pandemic. 

Different results with respect to the relationship between epidemiological data and the volume of activity on Twitter may be partly explained by a diverse use of this platform in different contexts (i.e., in some countries, Twitter is widely used in the general population; in other countries, it is mostly used for professional reasons) but also its use by public bodies for communication about the emergency.

Differently from the volume of posts, findings from the present study show that emotions were related to the epidemiological trend, hence providing useful insights into the coping strategies possibly adopted by the community to deal with the challenging and changeable situations, and their interplay with the communication strategy implemented. Fear and trust were the prevailing emotions detected. 

Recent studies conducted in other countries found a prevalence of negative sentiments during the first phase of the pandemic, which then shifted from negative to neutral sentiments, and even to positive attitudes once the community began to cope with the emergency [6,27]. Results from the present study confirm that trust increases while fear decreases over time. 

Fear is a typical sign of psychological distress arising as the first response to disaster and emergency situations that have been well-documented in the literature [44,45], and it is related to the uncertainty about the emerging health threat, the risk of contagion, and the impact of the outbreak on everyday life. A general sense of feeling lost is usually reflected in the frantic search for information, as a recent study demonstrated using SM analysis [6]. This feeling was very evident in the present study given the high volume of conversations about “virus and Pandemic impact” found. Fear might also have been amplified by the imposition of unprecedented restrictive measures [46], as demonstrated by peaks in tweets expressing fear at the beginning of the outbreak and when the national lockdown was declared. 

Properly addressing the widespread panic and fear is of paramount importance during a health emergency, as a preventive measure to mitigate the negative effects on mental health or the propensity towards risky behaviours. 

Indeed, a significant negative correlation was found between the percentage of tweets expressing fear and the frequency of bigrams related to “information seeking”. Information avoidance may mean that the public does not seek information on how to adapt to challenging situations. Policymakers would certainly benefit from real-time monitoring of information avoidance, (non) adherence to recommendations, and the dissemination of misleading or dangerous misinformation [47,48], as they may immediately plan counteracting strategies.

Again, from emotion analysis, this study also highlights that although the level of trust gradually increased, it decreased when quarantine was introduced, when it was extended, and when it was nearing the end. This confirms previous evidence that trust is generally much more influenced by key events and people’s reactions since it is closely connected to the perception of risk [49]. Moreover, during the first phase of the epidemic, trust negatively correlated with tweets about “Preventive measures”, perhaps reflecting the degree of disappointment towards the restrictive measures themselves, in line with findings from previous emergency situations [28,50]. Again, like in market research, having real-time knowledge about feelings associated with a specific theme may help develop strategies to reverse the perception with respect to a specific theme.

Some studies exploring web-based and SM activities linked to resilience and preparedness for COVID-19 and the preventive measures adopted have already been published, mostly reporting findings from China [6,51,52], while evidence from western countries is still limited. Supporting the initial hypothesis about the potential of tweets to reveal insights into people’s behaviour, in the present study a significant correlation was found between low levels of trust and the percentage of people who were found to violate the restrictions imposed by the law. 

While the present study does not provide definite conclusions about that specific finding, possible explanations could be the occurrence of shocking events or events with a high content of uncertainty, which lead to a decrease in the level of general trust (reflected also in the Twitter community), pushing people directly to a lower level of adherence to norms, or also indirectly impacting compliance with restrictive measures, through a higher propensity to share “fake news”.

*Limitations and future research*. We are aware that the Twitter community is not representative of the general population, and thus linking information gathered from Twitter to objective behavioural measures (e.g., the number of violations reported) is not sufficient to generalize these findings to those outside the Twitter community. 

Secondly, the authors assumed that tweets posted in Italian described reactions to the COVID-19 epidemic in Italy; however, some of those who tweeted may not have been actual residents in Italy, although there is a relatively limited proportion of people speaking Italian outside the country [53]. The language criteria were chosen rather than the extraction of geo-tagged tweets because of the limited availability and reliability of information about the location from tweets or metadata from other accounts. 

Thirdly, the study lacked an analysis accounting for geographical variability. Indeed, since the impact of the epidemic largely varied over regions, behaviours and social reactions were also likely to have varied as well. Further studies are needed to address this issue.

Finally, the main themes were selected based on a semantic analysis rather than on topic modelling, and comparisons between the two approaches are warranted in future studies. 

Future research should address the dynamics within the network of users’ community, also considering the role of influencers and their impact on the informational flow. Further studies could also explore the variability of the context in order to identify which variables are more strictly correlated to resilience and virtuous behaviours [42,54], to compare different settings, and even the various restrictive measures applied. Comparisons with similar data from subsequent waves of the COVID-19 pandemic are of paramount importance to understand the usefulness of data gathered from SM during different phases of health emergencies. Finally, a deeper analysis of the dynamics related to accounts of public interest (i.e., verified) and influencers is warranted in order to understand who plays the key role in the flow of information shared.

## 5. Conclusions

The present study contributes to the understanding of the reactions of communities during the COVID-19 pandemic on the basis of Twitter data and focused on the initial phase of the public health emergency when most of the reactions to the health emergency are spread and at the most critical time from a communication perspective. 

As in the previous context [55], results from the present study may aid in disclosing public response to inform government action and health policies. Adding to previous evidence, the study also highlights correlations between data retrieved from SM with Italian national data both related to the epidemiological curve and reports on the violation of restrictions, thus emphasizing the utility of integrating data from SM within existing pandemic management plans aimed at monitoring citizens’ resilience and awareness during prolonged emergency situations. 

Results from the present study offer insights into using data from SM to help develop decision-supporting systems and communication of public policymakers. 

In particular, findings support the use of SM to detect emerging themes and associated emotions, offering real-time knowledge that can be used to plan managerial changes according to trending themes but also to implement actions to create a positive outlook.

The development of a decision support system based on real-time surveillance of themes and emotions shared on SM offers knowledge for a prompt understanding of reactions and thus various behaviours. Such a system can help identify trending topic and those around which the discourse is declining, enable planning actions and organize services. Similarly, understanding emotions associated with the topics discussed helps promoting communication accordingly; this would increase the effectiveness of the messages and improve communication strategies for public health purposes. 

Indeed, such a surveillance system may support real-time management in the case of future emergency situations as well, even minor ones (e.g., recent monkeypox cases or the hepatitis cases).

## Figures and Tables

**Figure 1 ijerph-19-07785-f001:**
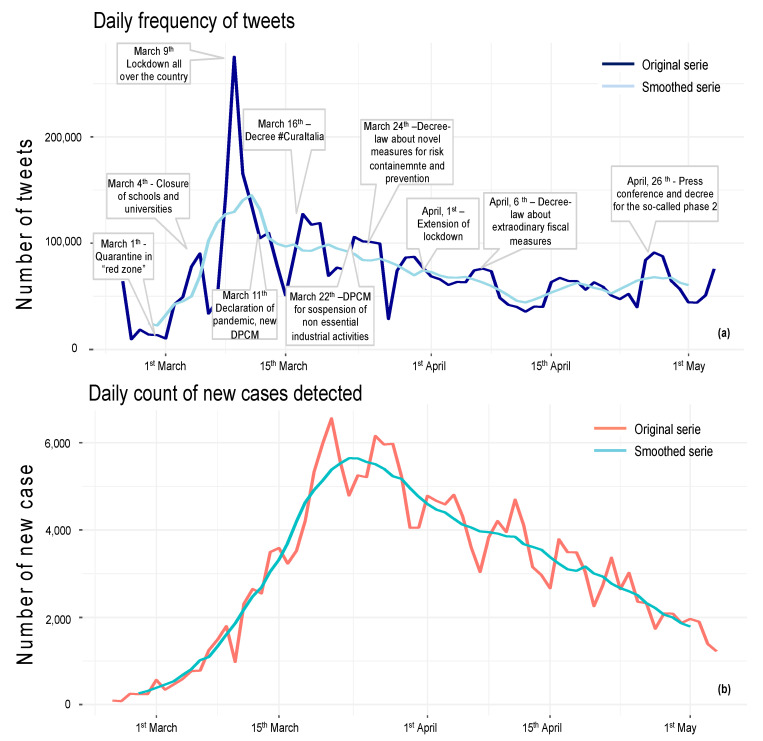
Daily counts of tweets and daily counts of newly detected COVID-19 cases. (**a**) Daily counts of tweets posted throughout the time frame with notes about national key events. (**b**) Timeline of newly detected cases. In all panels, national key events are highlighted, and the original time-series of row count and the smoothed series were obtained using simple moving average (MA) based on 7 days to remove peak and the “weekend effect”.

**Figure 2 ijerph-19-07785-f002:**
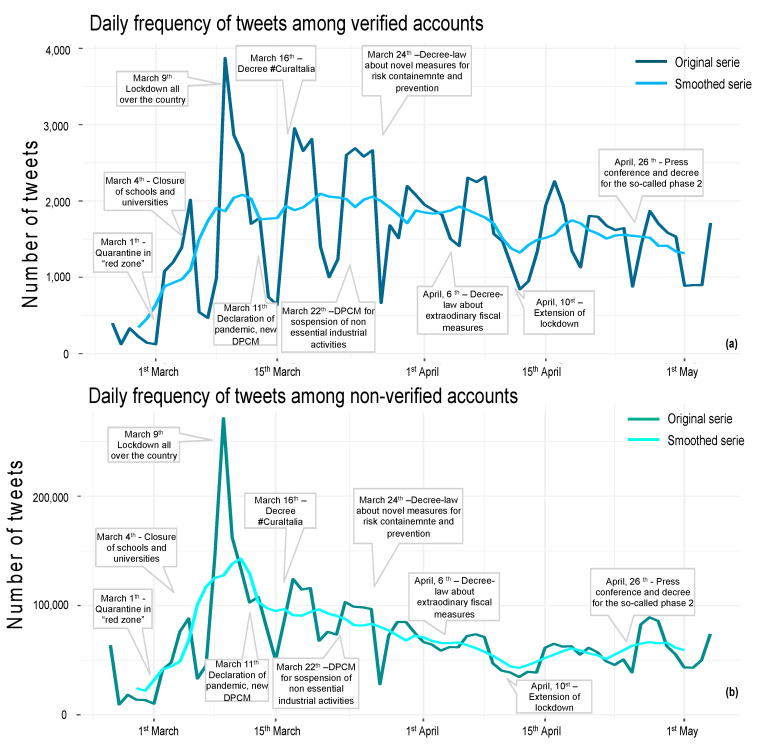
Daily counts of tweets. (**a**) Daily counts of tweets (and retweets) posted throughout the time frame among verified accounts. (**b**) Daily counts of tweets (and retweets) posted throughout the time frame among non-verified accounts. In all panels, national key events are highlighted, and the original time-series of row count and the smoothed series were obtained using simple moving average (MA) based on 7 days to remove peak and the “weekend effect”.

**Figure 3 ijerph-19-07785-f003:**
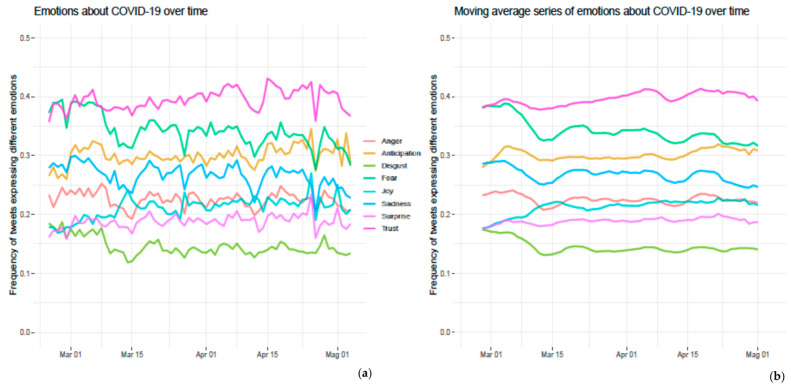
Timeline of emotions over time. (**a**) Daily frequency of tweets expressing different emotions; (**b**) smoothed series obtained using simple moving average (MA) based on 7 days to remove peak and the “weekend effect”.

**Figure 4 ijerph-19-07785-f004:**
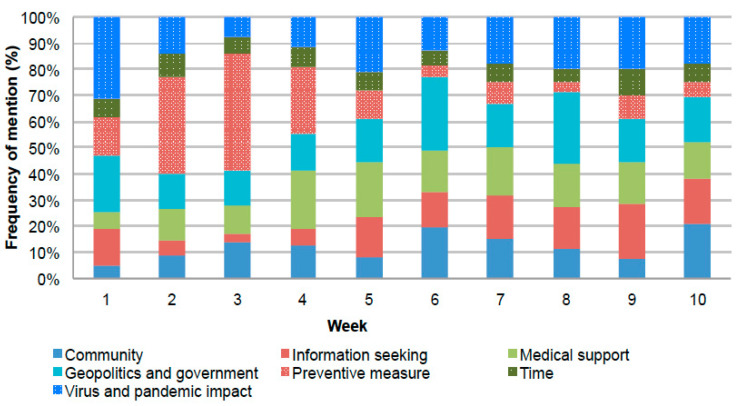
Frequency of mention of themes over weeks.

**Table 1 ijerph-19-07785-t001:** Results from the semantic analysis, theme identified, top bigrams associated and overall frequency of mentions throughout the study period.

Theme	Top Bigrams	%
Preventive measures	Zona rossa; stare casa; state casa; Italia zona; porte chiuse; zone rosse; scuole chiuse; restare a casa; rimanete a casa; zona protetta	21.5%
Geopolitics and government	Tutta Italia; protezione civile; segretario generale; cessate fuoco; fuoco globale; globale firma; cura Italia; appello segretario; task force; presidente consiglio	17.3%
Medical support	Terapia intensiva; medici infermieri; prima linea; operatori sanitari; personale sanitario; posti letto; sistema sanitario; sanità pubblica; pronto soccorso; test sierologici	15.5%
Virus and pandemic	Nuovi casi; emergenza coronavirus; casi positivi; emergenza sanitaria; contagio prego; coronavirus COVID; corona virus; emergenza epidemia; prendere COVID; virus COVID	14.1%
Community	Buona Pasqua; andrà bene; fare spesa; raccolta fondi; migliaia di persone; persone mondo; sostenendo appello; mondo sostenendo; scuole università; forza Italia	12.4%
Information seeking	Primo caso; conferenza stampa; ultime notizie; cosa fare; diventando virale; live coronavirus; coronavirus ultime; informazioni aggiornamenti; notizie diretta	11.3%
Time	Fino aprile; ogni giorno; fino marzo; due mesi; fino maggio; due settimane; ultime ore; quest’anno; metà marzo; fino luglio	7.9%

## Data Availability

Not applicable.

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
