# Peer review of "How Italy Tweeted about COVID-19: Detecting Reactions to the Pandemic from Social Media"

_ijerph, 2022, doi:10.3390/ijerph19137785_

Round 1
Reviewer 1 Report
First of all, congratulations to the authors for this very complicated and challenging bet. Social networks have a fundamental weight in our daily lives and, of course, they did during the pandemic. It was our way of communicating.
Below are some comments on minor issues that I have detected in the document.
1. In data analysis I have seen that non-parametric tests were used, but I have not seen the justification for their use. Did you test for normality?
2. In the same section, the significance level and alpha used have not been clarified.
3. The graphs (Figure 1) do not have the legend included. That is, It cannot know what each line means. The words inside the boxes cannot be read clearly.
4. In the results, the p-value should be in italics.
5. In the same section, I have seen that it always indicates the value of p, but the value of the statistic should also be included.
6. I haven't seen the design explained explicitly. I can think that it is a descriptive design, but I think that it should be clear.
7. Sometimes the first person is used excessively. It must have a neutral language.
8. In some points I think the document has subsections that have not been differentiated from the text, so it seems that the sentence is incomplete. For example, lines 256 and 358.
Thank you very much,
Reviewer 2 Report
This paper provides the analysis of the reactions of 385 communities during the COVID-19 pandemic on the basis of Twitter data. Our results 392 offer insights into using data from social-media to help develop communication of public policy 393 makers. The following suggestion for the authors to consider.
1. Figure 1 is rather vague, it can be enlarged and presented more clearly.
2. The research methods are rather conventional. The analytical methods can be more specific. The lack of the details about the analytical process weakens the paper's significance and novelty.
3. The conclusion should include some detailed solutions.
Reviewer 3 Report
Thank you for the opportunity to review the paper titled ‘How Italy tweeted about COVID-19: detecting reactions to the pandemic from social media’.
I offer my detailed comments regarding this paper below.
Introduction
The introduction has an overall logical flow. However, I am not convinced by the link made to Deming’s framework in justifying the use of social media for detecting attitudes and behaviors and informing policy. The framework comes as a surprise to the reader and its relevance to the paper’s topic is rather descriptive and speculative. I would encourage the authors to either provide a stronger (and more analytical) justification for the relevance of this framework to their topic or remove this framework altogether.
Materials and methods
Setting: Here the authors should explain what is interesting about Italy as a context of study in terms of social media (e.g., social media usage during COVID-19 in Italy increased, Twitter is a main platform for Italians to obtain information from etc.) or perhaps something else (?) and not just merely rely on reporting widely available descriptive COVID-19 data.
The authors also need to more explicitly justify their choice of data collection period. As a reader, I cannot help but wonder why the authors chose to focus on such a short period of time and whether it would have been useful to do a longitudinal study – for example, comparing 2020 to 2022 data? In other words, why is this period deemed crucial in understanding COVID-19 behaviors and attitudes?
Data sources and pre-processing: For purposes of clarity, I think that here you should add a sentence at the start that you have used two datasets: Twitter data and data from the Italian Department of Civil Protection.
Data analysis: Please elaborate more at the end about coding disagreements and how the third coder resolved these.
Please give an example of ‘pairs of adjacent words (bigrams)’ after this sentence: ‘To explore the most common themes in the tweets, a semantic analysis of the most frequent pairs of adjacent words (bigrams) was performed.’
The authors discuss six pre-defined themes from past research on COVID-19. Please specify whether these are the themes that were used in the coding process discussed in the previous paragraph and if so, integrate these paragraphs better.
Results
This sentence ‘A total of 4,988,255 Twitter posts (1,256,890 tweets and 3,731,365 retweets) referred 175 to conversations about COVID-19 were shared in Italian from February 25th 2020 to May 176 4th 2020’ should be moved to a relevant sub-section in the ‘Materials and methods’ section. A possible suitable place would be at the end of paragraph one in ‘Data sources and pre-processing’. You should also remove from the sentence here the data collection period, because it becomes repetitive.
The presentation of your findings is otherwise fine.
Discussion
You do not need to discuss your study’s context (Italy) again here (paragraph 2), provided that a stronger justification is provided in the paper early on.
Here you need to more clearly convey your contributions to theory/existing research – where do you position within the current studies that you briefly discuss in the sentence starting ‘Recent studies…’?
Your discussion section is difficult to follow. In the fourth paragraph starting ‘The main results are...’ you outline your key findings well and it would be useful to follow these up with four main paragraphs on theoretical and managerial implications, rather than multiple somewhat unfocused paragraphs.
You also need to more clearly separate between your theoretical and managerial implications.
Again, I am not convinced by the relevance and contribution to the Deming’s framework in the final paragraph.
Limitations
In my opinion, your data collection period is a big limitation to your study. I encourage you to discuss it in your limitations and put forward ideas on how future studies can overcome similar research limitations.
Conclusion
In your conclusion, you should touch upon the relevance of your research in future, given that we are now in 2022 and the impact of the pandemic is a lot less prevalent than it was back in 2020.
Finally, I would encourage you to have your paper professionally proof-read.
Best of luck.
Round 2
Reviewer 3 Report
Thank you for your extensive revisions. I am happy to accept the paper in its current form and only I encourage the authors to conduct a final proofreading of their work.